# Is Cell-Free DNA Testing in Pancreatic Ductal Adenocarcinoma Ready for Prime Time?

**DOI:** 10.3390/cancers14143453

**Published:** 2022-07-15

**Authors:** Ankur Sheel, Sarah Addison, Surya Pratik Nuguru, Ashish Manne

**Affiliations:** 1Department of Internal Medicine, The Ohio State University College of Medicine, Columbus, OH 432120, USA; ankur.sheel@osumc.edu; 2School of Medicine, The Ohio State University, Columbus, OH 432120, USA; sarah.addison@osumc.edu; 3Department of Internal Medicine, Kamineni Academy of Medical Sciences and Research Center, Hyderabad 500012, India; suryanugooru@gmail.com; 4Department of Internal Medicine, Division of Medical Oncology at the Arthur G. James Cancer Hospital and Richard J. Solove Research Institute, The Ohio State University Comprehensive Cancer Center, Columbus, OH 43210, USA

**Keywords:** cell-free DNA, pancreatic ductal adenocarcinoma, epigenetic markers, liquid biopsy, somatic mutations, methylation markers

## Abstract

**Simple Summary:**

Pancreatic cancer is a deadly cancer with limited treatment options. It is often detected in most people at stages where cure is not possible. There is no good test to know if a person will respond to treatment or if there is any disease beyond what can be seen by available imaging tests. Genetic material from the tumor is expected to float in the blood. Studying the alterations in the genetic material could help detect the tumor early, give an idea about its aggressiveness and response to available treatments, and facilitate the discovery of newer therapies. The focus of the studies so far has been on only one kind of genetic aberration, mutations, which has not given us great results. There is a need to explore another type of change known as methylation that could hold answers for managing pancreatic cancers better.

**Abstract:**

Cell-free DNA (cfDNA) testing currently does not have a significant role in PDA management: it is insufficient to diagnose PDA, and its use is primarily restricted to identifying targetable mutations (if tissue is insufficient or unavailable). cfDNA testing has the potential to address critical needs in PDA management, such as pre-operative risk stratification (POR), prognostication, and predicting (and monitoring) treatment response. Prior studies have focused primarily on somatic mutations, specifically *KRAS* variants, and have shown limited success in addressing prognosis and POR. Recent studies have demonstrated the importance of other less prevalent mutations (*ERBB2* and *TP53*), but no studies have provided reliable mutation panels for clinical use. Methylation aberrations in cfDNA (epigenetic markers) in PDA have been relatively less explored. However, early evidence has suggested they offer diagnostic and, to some extent, prognostic value. The inclusion of epigenetic markers of cfDNA adds another dimension to genomic testing and may open new therapeutic avenues beyond addressing critical areas of need in PDA treatment. For cfDNA to substantially influence PDA management, concerted efforts are required to include less frequent mutations and epigenetic markers. Furthermore, relying on *KRAS* mutations for PDA management will always be inadequate.

## 1. Introduction

Pancreatic ductal adenocarcinoma (PDA) is a fatal cancer with dismal survival. The 5-year relative survival is 11% for all-stage PDA: 42% in localized tumors and 3% in metastatic tumors [1]. In the United States, PDA is expected to be tenth in incidence (>62,000) and third in mortality (approximately 50,000) among other malignancies in 2022 [1]. A lack of reliable screening protocols, the high frequency of diagnosis at advanced stages, the need for debilitating surgery with a very high rate of recurrence, the limited systemic treatment options (adjuvant therapy (AT), neoadjuvant therapy (NAT), and palliative treatment (PT)), and the absence of reliable biomarkers contribute to poor outcomes in PDA [2]. The current status of PDA management and the areas requiring critical attention, beyond the development of novel therapeutic agents, are discussed in this section.

### 1.1. Current Management of PDA: Treatment and Disease Monitoring

Surgical resection is offered for early-stage tumors (both resectable (R-PDA) and borderline resectable (BR-PDA) if feasible) as a curative treatment [3]. Unfortunately, 80–85% of patients present with advanced disease (aPDA), including metastatic (mPDA), and locally advanced (LA-PDA) tumors [4]. The recurrence rates after successful resection are very high (80%), and the median long-term survival (>10 years) is only 4% [5]. Chemotherapy (FOLFIRINOX or gemcitabine/nab-paclitaxel (G-NP)) remains the main component of PDA systemic management (for NAT, AT, or PT) [6,7].

The current standard of care uses a combination of imaging and carbohydrate antigen 19-9 (CA 19-9) levels to monitor treatment response in patients receiving systemic therapy and to detect recurrence during surveillance [8]. This approach is limited by not only the poor sensitivity of CA19-9 but also the latency and evolving diagnostic uncertainty surrounding radiographic changes associated with PDA [9,10]. Moreover, CA19-9 is an unreliable biomarker because it is elevated in benign diseases (such as acute cholangitis, cirrhosis, and cholestatic diseases) and biliary obstruction from tumors, and shows an absence of elevation in 5–10% of PDA cases [11,12,13]. Therefore, an urgent need exists for new techniques or biomarkers to aid in diagnosis, staging, and therapeutic decision-making for PDA. The newest first-line therapy for aPDA was approved more than a decade ago. Newer targets and agents to treat PDA are beyond the scope of this review. Other ways to improve outcomes are discussed below.

### 1.2. Critical Needs in Pancreatic Adenocarcinoma Management

A comprehensive approach to PDA should start with developing more effective tools to aid in (a) diagnosis (and screening); (b) prognostication to anticipate the outcomes at diagnosis, irrespective of disease stage; (c) pre-operative risk stratification (POR); and (d) predicting (choosing an appropriate chemotherapy combination, G-NP vs. FOLFIRINOX) and monitoring treatment response. Traditional genomic testing and personalized approaches have not revolutionized PDA management as expected. The gold standard method of diagnosis (particularly for non-metastatic PDAs and mPDA with inaccessible metastatic disease) is endoscopic ultrasound-guided fine-needle aspiration (EUS-FNA) [8]. This approach provides a limited number of cells for cytological analysis and is scarcely sufficient for diagnosis. It often fails to provide adequate tumor tissue to perform genomic testing, including next-generation sequencing (NGS). Alternatives to tissue testing, such as cell-free DNA (cfDNA) testing in blood or liquid biopsies, have garnered much public attention as a non-invasive technique that may aid in diagnosis, staging, and therapeutic decision-making for malignancies [14].

cfDNA includes normal cell DNA, tumor cell DNA (ctDNA), circulating tumor cells (CTCs), and exosomal DNA, which can be detected in non-tissue sources such as blood, pancreatic juice, and bile [15]. Genetic aberrations in cfDNA from the blood of patients with cancer, specifically somatic mutations, are accessible to treating physicians, but their use is currently limited to the identification of targetable mutations cfDNA; testing in its current form cannot serve as a method for biomarker analysis. However, methylation changes or epigenetic markers in blood cfDNA are slowly gaining prominence. Most studies on cfDNA in PDA have involved detection and/or quantitative analysis of *KRAS* variants (mut-*KRAS*) in blood cfDNA. Among the cfDNA epigenetic markers, changes in methylation patterns have been more studied than histone modifications. In this review, cfDNA denotes cfDNA from the blood unless another source (bile or pancreatic juice) is specified. The current roles of somatic mutations (mostly mut-*KRAS* variants) and methylation changes detected in the cfDNA in PDA management are discussed in the review, with a focus on the predictive, prognostic, and POR value. The diagnostic value of epigenetic markers is discussed in depth, because of the limited relevant information in the literature.

## 2. Detection of cfDNA in PDA

Tumor cells in the cardiovascular system were first described in 1869, when Australian pathologist Thomas Ashworth observed cells in a post-mortem blood sample that were morphologically identical to those in the >30 subcutaneous tumors found in the same patient [16]. This seminal work suggested that the cells must pass through the circulatory system for tumors to metastasize. This was the first written report suggesting that cells from a tumor could be found within the circulatory system. Almost a century later, in 1955, the surgeon H.C. Engell corroborated Ashworth’s observations and observed morphologically identical cells in the bloodstream of a patient with colorectal cancer [17]. The prognostic and diagnostic value of these cells, termed CTCs, was unclear at the time. The first report of nucleic acids found in circulating plasma was by Mandel and Metais in 1948 [18]. Over the past decade, research interest has been renewed in this circulating material, because studies have demonstrated that these markers can aid in the diagnosis of a variety of cancers.

Another study in 1977 demonstrated a correlation between patients with malignancy and the absolute concentration of cfDNA in the serum [19]. In that study, Leon et al. demonstrated elevated concentrations of detectable cfDNA in serum from patients with lymphoma, lung, ovary, uterus, and cervical tumors, which decreased after disease-directed radiation therapy. It was not until 1994 that the first point mutation was detected in cfDNA [20]. With advances in sequencing technologies, the first somatic point mutation was identified in the *NRAS* gene, isolated from cfDNA in patients with myelodysplastic syndrome and acute myeloid leukemia. As sequencing technologies have advanced, both epigenetic modifications and RNA components of cfDNA can be identified [21]. The mechanism of cfDNA release by cells into the circulation remains not fully understood. A set of preliminary studies using samples from patients with colorectal cancer have postulated that cfDNA arises from products associated with necrosis, active secretion of DNA (exosomes and DNA–lipoprotein complexes), and apoptosis, as the sizes of sequenced cfDNA fragments correspond to those after DNA cleavage events associated with necrosis and apoptosis, and the sizes of fragments found in exosomes [22].

The mere presence and concentration of cfDNA are insufficient to guide clinical decisions, because many other conditions beyond cancer alter levels of cfDNA, including inflammatory conditions, cerebrovascular accidents, exercise, smoking, and trauma [23]. Because PDAs are notoriously difficult to diagnose via EUS-FNA, the detection of cfDNA provides an alternative method for PDA diagnosis. Numerous studies have demonstrated that cfDNA is detectable in the blood of patients with PDA across various stages of the disease, through methods including traditional polymerase chain reaction (PCR), pre-developed assay reagents (PDAR), PDAR-restriction fragment length polymorphism (PDAR-RFLP), digital droplet PCR (ddPCR), beads, emulsion, amplification, and magnetics PCR (BEAM-PCR), and NGS [14]. The first study to detect cfDNA in patients with PDA used PCR to amplify the *KRAS* oncogene in three patients and subsequently used traditional Sanger sequencing to identify mutant variations in the *KRAS* oncogenes (mut-*KRAS*) [24]. Studies have demonstrated that the detection of genetic aberrations in cfDNA is possible; more importantly, they have highlighted that these abnormalities are highly representative of those found in the primary tumor, thus suggesting that the somatic mutations identified in cfDNA are concordant with those found in tumor tissue [25,26].

Studies in the literature on cfDNA somatic mutations (Appendix A) can be broadly divided into either total cfDNA (non-specific) mutations or specific mutations such as *KRAS*, *TP53*, and *CDK2NA*. The parameter of interest in these studies is the detection of mutation(s) or quantitative analysis of detected mutations or the mutant allelic fraction (MAF, a measure of the percentage of mutant alleles among all alleles in any given sample). Serial cfDNA testing studies to monitor treatment response, disease burden, or outcomes typically involved following one of these parameters. Similarly, methylation assays may be non-specific (such as total methylation levels) or specific to promoters (hypo- or hypermethylated), differentially methylated regions (DMRs), or differentially methylated CpG sites (DMPs). The concentration or degree of methylation in a promoter, measured as the methylation index (MI, methylated copy number/methylated copy number + unmethylated copy number), has been assessed in some studies. In this review, the available evidence is divided into sections according to current clinical needs to enable better understanding of the role of cfDNA testing.

### 2.1. cfDNA Testing in PDA, beyond KRAS

In an abstract presented at GI ASCO 2021 of a study including 1,009 patients from the database of Foundation One Medicine, 61% (613/1009) of the patients had at least one somatic alteration detected in cfDNA through comprehensive genomic profiling [26]. Among the detected alterations, *TP53* was the most frequently altered (55%), followed by *KRAS* (40%) and *CDK2NA* (6.5%). The other mutations in the top ten were *ATM* (2.3%), *PIK3CA* (2.2%), *PTEN*, *TERT*, *NF1*, *JAK2*, and *GNAS*. The frequency of the last five mutations in this list was below 1.5%. The frequency of the altered mutations in the blood (cfDNA) correlated with the frequency of mutations detected in tissue in the available data from 81 patients [25]. *KRAS*, *TP53*, and *CDK2NA* were the top three mutated genes in tissue and blood. In a similar study (*n* = 23), the prevalence of detected mutations was higher, at 78% for both *TP53* and *KRAS* [27]. This study used an NGS panel from a different company (Guardant Health, Redwood City, CA, USA) than the previous study. A study using ddPCR for a five-gene panel (*n* = 188) identified other mutations: *BRCA2* (11.7%), *KDR* (13.8%), *EGFR* (13.3%), *ERBB2* exon17 (13.3%), and *ERBB2* exon27 (6.4%) [28]. The mut-*KRAS* rate was 72.3 % (G12V, G12D, and G12R rates were 34.5%, 52.1%, and 9%, respectively) in the same study.

A large sequencing-based study in 21,807 patients (25,578 blood samples) with treated late-stage cancers demonstrated the abundance of cfDNA in blood samples from patients with malignancy [29]. Alterations tested include single nucleotide variants, copy number variations, insertions and deletions, and fusions (from the Guardant Health panel). This comprehensive study examined >50 cancer types and had 1112 (4%) cfDNA samples from patients with PDA. Somatic alterations (total or non-specific) were detected in approximately 80% of the tested PDA samples; however, much larger studies might be required to understand their roles. In a similar but smaller study (*n* = 410 and 155 PDA), cfDNA was detected in >80% of aPDAs and 48% of localized PDAs (although further details on tumor stages were not reported) [30].

The concordance of the genetic alterations between tissue or EUS-FNA samples and cfDNA in PDA was studied in a meta-analysis (14 studies with 369 patients) published in 2019 [31]. Most studies included in this meta-analysis examined mut-*KRAS* (as expected), except for three studies that used multigene panels. Two studies analyzed CTCs, and the remainder performed cfDNA testing. Overall, the sensitivity and specificity of liquid biopsy to diagnose PDA compared with tissue specimens in this meta-analysis were 70% and 86%, respectively. Interestingly, when mut-*KRAS* alone was analyzed, the sensitivity decreased to 65%, but the specificity increased to 91%. The concordance rate in the study was 32%, while the mutations exclusively detected in the blood and tissue samples were just 30% and 38%, respectively. Because cfDNA contains genetic material from various sources (e.g., tumor cells, normal cells, and exosomes), its profiles are more likely to represent genetic variability and provide greater genomic insight into tumor heterogeneity than direct tissue sampling [32]. Although the low concordance in PDA may be disappointing, PDA is a cancer in which tissue samples are difficult to obtain. cfDNA testing can identify 62% of mutations (alterations), and 30% of them are different from those detected in tissues. Interestingly, a concordance rate of >75% has been found in studies of mut-*KRAS* alone [33,34]. Therefore, cfDNA testing can serve as an alternative in PDA patients.

The important takeaway from these PDA cfDNA studies is that mut-*KRAS* is detected in 40–50% of samples, but other less prevalent mutations make up the other 30–40%, and targetable mutations are rare in PDA. Therefore, to make cfDNA testing relevant in PDA, a concerted effort is needed to expand beyond mut-*KRAS* testing. Studying the clinical value of other mutations for prognosis or POR is difficult because large studies are required to indicate correlations. One strategy involves adding another layer to testing by using epigenetic markers.

Contrary to the common assumption that cfDNA can be detected only in advanced PDAs, many studies in the literature have reported its detection in early-stage tumors [35,36], thus not only improving POR (discussed below) but also aiding in diagnosis/screening (briefly discussed in Appendix A). In current clinical practice, diagnosis cannot be made based on cfDNA testing alone outside experimental settings, but it can provide support in rare cases in which the histological diagnosis is inconclusive and the suspicion for PDA is very high. 

### 2.2. cfDNA Epigenetic Markers

Methylation in the promoter regions of genes is a major regulator of gene activation and inactivation [37]. Malignancy is often associated with hypermethylation of tumor suppressor genes, thus resulting in genetic inactivation, and hypomethylation of oncogenes, thus resulting in genetic activation [37]. Advances in sequencing technologies have also allowed for the identification of hypermethylation and hypomethylation patterns in cfDNA [14]. cfDNA epigenetic markers can be used for diagnostic purposes, as suggested by several retrospective studies summarized in Table 1 below. In some studies, cfDNA epigenetic markers from PDA patients have been compared with those from healthy controls (HC) or with those having benign diseases, such as cysts, chronic pancreatitis (CP), or both, to derive diagnostic markers.

In an early study, differentially expressed methylated cfDNA patterns were compared in patients with PDA (all stages) and normal patients [38]. The authors of the study curated a panel of five genes—*CCND2*, *PLAU*, *SOCS1*, *THBS*, and *VHL*—enabling differentiation between malignant and normal samples (HC) with a sensitivity and specificity of 76% and 59%, respectively. The methylation frequency of the genes in the panel was lower in PDA cfDNA than HC. The authors then curated a panel of cfDNA epigenetic markers that successfully discriminated between PDA and CP with a sensitivity and specificity of 91.2% and 90.8%, respectively [39]. Interestingly, methylation of *NPTX2* was able to differentiate PDA from CP but not from HC [54,55]. In addition, greater *p16* promoter methylation has been observed in PDA than HC. A genome-wide methylation study has demonstrated that hypermethylation of the *SST* gene can be detected in cfDNA across all stages of PDA development, with a sensitivity of 89% for PDA diagnosis [46]. Recently, multiple panels with reliable diagnostic accuracy have been reported for distinguishing PDA from HC or CP [42,44,45,56]. In one study using a 28-gene promoter panel, the number of methylated genes in the PDA group (8.41 vs. 4.74, *p* < 0.001) was significantly higher than that in the non-cancer group (HC + CP + acute pancreatitis).

Advances in methylome sequencing techniques have led to a dramatic increase in studies examining promoter hypermethylation in cfDNA. A recent study generated a panel of 51 features including both 5-methylcytosine (5mC) and 5-hydroxymethylcytosine (5hmC) methylation patterns that can distinguish between patients with PDA and healthy individuals, with a sensitivity of 93.8%, specificity of 95.5%, and AUC of 0.997 [2]. The compound 5hmC, a derivative of 5mC, is a distinct epigenetic marker of transcriptional regulation of normal differentiation [57]. Its global loss and re-distribution are common features in various cancers, including PDA [58]. Individually, the sensitivity and specificity of 5mC alone are better than those of 5hmC. Interestingly, gene set enrichment analysis (GSEA) of 794 differentially hydroxymethylated genes (differential 5hmC) identified in cfDNA from patients with PDA demonstrated a similar genetic pattern to those commonly dysregulated after *KRAS* activation and *TP53* inhibition in primary tumor tissue [43]. This finding suggests that these differentially methylated genes are likely to originate from the tumor itself. Most of the genes identified are associated with pancreatic development or function (*GATA4*, *GATA6*, *PROX1*, *ONECUT1*, and *MEIS2*), and malignant transformation (*YAP1*, *TEAD1*, *PROX1*, and *IGF1*). A predication model developed by using the top 65% of genes with high 5hmC variation has shown an impressive AUC of 0.92–0.94. Additionally, a model developed from 5hmC data on tissues has been reported to have an AUC of 0.88 when tested in cfDNA.

EpiPanGI Dx is another NGS-based methylome panel developed to aid in the diagnosis of PDA [50]. To develop this tool, researchers compared DMRs between tumor tissue and adjacent normal tissue, then cross-referenced these regions with methylated regions that were differentially identified in cfDNA from patients with malignancy and healthy patients. Overall, this assay has a predictive value of approximately 85% for the diagnosis of PDAs. Similar studies have provided other DMPs and DMRs for the diagnosis of PDA [59,60,61]. One group has demonstrated that methylation levels in terms of the MI of four genes, secreted protein acidic and rich in cysteine (*SPARC*), ubiquitin carboxy-terminal hydrolase L1 (*UCHL1*), neuronal pentraxin 2 (*NPTX2*), and proenkephalin (*PENK*), can distinguish between HC, CP, and PDA [47]. They have reported several interesting results (Table 1). The MI of all genes was higher in abnormal patients (CP and PDA) than HC, and that of *SPARC* distinguished early-stage PDA from CP, and thus may be useful for screening patients.

Shinjo et al. developed a panel of five markers (*ADAMTS1*, *HOXA1*, *PCDH10*, *SEMA5A*, and *SPSB4*) through tissue-based DNA methylation microarray analysis of PDA (*n* = 37), which has shown impressive diagnostic accuracy in The Cancer Genome Atlas (TCGA) dataset and an independent cohort (*n* = 146) [48]. For cfDNA, methylated DNA was identified with methyl-CpG binding (MBD) proteins, followed by ddPCR. The results of cfDNA testing were not encouraging (47 PDA and 14 HC). One of the five markers was positive in 49% (23/47) of patients with PDA, with a sensitivity and specificity of 49% and 86%, respectively. When mut-*KRAS* was added to the panel, the sensitivity improved to 69% but the specificity remained the same. Comparison of these markers in paired tissue and cfDNA samples (*n* = 29) indicated that all tissues had mut-*KRAS* and one of the five markers, whereas 19/29 (66%) cfDNA specimens had either the mut-*KRAS* or one of the markers; moreover, genome wide sequencing showed an 80% overlap of methylated regions and a significant correlation (R = 0.97) between cfDNA and tissue DNA methylation profiles. This study demonstrated the advantage of adding somatic mutation to epigenetic markers from all available sources in the diagnosis of PDA, and the reliable concordance of methylation profiles between cfDNA and tissues. 

Overall, a cfDNA epigenetic marker can be a specific gene promoter or DMP/DMR or MI. The prevalence of some of these markers is as high as 90% in PDA and much lower in people without cancer (HC and CP), thus indicating their substantial advantages over somatic mutations [41,47]. Markers that can distinguish HC from PDA may be developed for screening, and those that can differentiate PDA from benign disease (cysts and CP) could be used in the diagnosis of suspected cases and surveillance. 

## 3. Prognostic Value of cfDNA in PDA

Biomarkers to forecast outcomes are needed to decide on appropriate treatment approaches, particularly in patients who present with aPDA and poor performance status secondary to complications of metastatic disease, including liver failure and pain. These markers may also aid in the precise staging of early-stage PDA on imaging (discussed below). Most cfDNA studies in PDA have involved detection and quantification of mut-*KRAS*, because their prevalence is higher than those of other known mutations. In the past 4–5 years, the prognostic value of epigenetic markers has also been reported, but more evidence on them is needed.

### 3.1. cfDNA KRAS Mutations

*KRAS* is the most frequently mutated driver of PDA detected in tissue [62]. Multiple studies have detected *KRAS* mutations in cfDNA from both blood/serum and biliary fluid in patients with PDA. The poor prognostic implications of finding mut-*KRAS* in the cfDNA of patients with PDA were first reported in 1999 [63]. This early study examined *KRAS* codon 12 mutations in 44 patients with PDA. The 6-month (7% vs. 41%) and 12-month (0% vs. 24%) survival rates were significantly lower in patients with mut-*KRAS* than wild-type *KRAS*. Studies examining the prognostic value of specific *KRAS* mutations from cfDNA are summarized in Table 2.

Almost 22 years later, more than 3000 published studies have examined the prognostic roles of *KRAS* mutations in cfDNA. Mut-*KRAS* detection in cfDNA is typically associated with advanced stages and more distant metastasis than early stages [64,66,67,71,79]. One study identified these mutations in just 8% of patients with R-PDA but in 60% of patients with metastasis [79]. The detection of mut-*KRAS* in aPDAs indicates poorer outcomes, as demonstrated by multiple studies [64,67,73,74,76,80,82]. A recent meta-analysis of 48 studies examined the prognostic role of mut-*KRAS* detection in cfDNA to provide a more comprehensive summary [83]. The study examined 3524 patients across all stages of the disease, before and after treatment (both surgical and pharmacologic). Mut-*KRAS* detection was profoundly associated with the PDA outcomes, irrespective of the stage (hazard ratio (HR) = 2.42, 95% CI: 1.95–2.99 for OS and HR = 2.46, 95% CI: 2.01–3.00 for PFS) and in advanced stages (HR = 2.51, 95% CI: 1.90–3.31). The absence of detectable mutations (*KRAS* negativity) after either surgical or pharmacological treatment in patients with prior mut-*KRAS* detection has been found to indicate favorable prognosis (HR = 2.46, 95% CI: 2.01–3.00). Additionally, researchers have empirically demonstrated that, when combined with CA19-9, the detection and concentration of mut-*KRAS* in cfDNA have greater prognostic value than those of CA19-9 alone across all stages (HR = 2.08; 95% CI: 1.20–3.63) [33].

In addition to detection, quantification of mut-*KRAS* can also confer prognostic value. Among patients with mut-*KRAS*, higher concentrations and higher MAF often correlate with aPDA [79,84]. A multivariate analysis of factors influencing outcomes (*n* = 77, 35 R-PDA, 36 mPDA, and 6 LA-PDA) demonstrated that mut-*KRAS* concentrations >0.165 copies/L are associated with poorer OS [71]. Poorer PFS and OS were observed in these patients in the same study after univariate analysis (UVA). In contrast, KRAS fractional abundance (>0.415%) was associated with PFS in UVA but not in OS. Similarly, in another study, a higher mutational load (MAF with a cutoff of 0.351%) was associated with poor PFS (175 days vs. 85 days, HR = 2, *p* = 0.05) and OS (310 days vs. 142 days, HR = 2.2, *p* = 0.02) [80]. That study even demonstrated that an increasing MAF over time is a poor prognostic factor. Mohan et al. showed that, along with MAF, copy number gains are also associated with poor outcomes [73].

All mutations associated with *KRAS* genes do not uniformly influence outcomes. The specificity of the codon and particular point mutations are also influential. Variants from codon 12, specifically G12V, are associated with poorer prognosis than other aberrations [34,65,67]. In aPDA, G12D, G12V, and high T-regulatory cells (Tregs) are associated with poor outcomes independently, but the outcomes are much poorer in patients with G12V and high Tregs [68]. Furthermore, patients with G12R, compared with other variants of *KRAS*, have longer OS (20.4 vs. 14.5 months, HR = 0.67 (95% confidence interval (CI): 0.47–0.93)). However, in that study, only 4% of the patients had ctDNA NGS [70]. The source of the DNA (ctDNA vs. exosomal DNA) in cfDNA also affects patient outcomes. Bernard et al. noted that exosomal mutation (>5%) alone or in combination with ctDNA is an indicator of poor prognosis [66]. Overall, mut-*KRAS* detection, quantification, and specificity (of the point mutation) are important influences on prognostic value.

### 3.2. cfDNA KRAS with Other Mutations in PDA

Although most studies have focused on mut-*KRAS*, some studies have demonstrated that the detection of other mutations also has prognostic value [75,77,85,86]. Some cfDNA-NGS studies have distinguished the value of mut-*KRAS* and other mutations. In these studies, the influence of the concentration of all mutations (including mut-*KRAS*) detected in cfDNA on patient outcomes was assessed. At the end of a 28-week study period, the survival rate was 54% in mutation-positive patients compared with 90% in mutation-negative patients in one study [75]. Stirjker et al. also showed that detection and a higher MAF of multiple mutations independently predict poor outcomes [77].

In a study with 127 patients with PDA (53/127, mPDA), mut-*KRAS* detection was not found to be significantly associated with clinicopathological factors or survival [85]. However, higher concentrations (>62 ng/mL) of cfDNA are also associated with adverse features, such as distant metastatic disease (median of 92 ng/mL (15–239) vs. 58.7 ng/mL (15–240), *p* = 0.01), vascular invasion (median of 78 ng/mL (15–240) vs. 63 ng/mL (15–239), *p* = 0.03), and poorer OS (3 months in >62 ng/mL vs. 11 months in ≤62 ng ng/mL, *p* = 0.22, HR = 2.6). Higher cfDNA levels were also found to be an independent risk factor in multivariate analysis (HR = 2.8 (95% CI: 1.8–4.6), *p* = 0.01). This study first quantified cfDNA by spectrophotometry and used PCR to identify mut-*KRAS* (codon 12). Similar findings were reported in a recent meta-analysis of 38 studies (*n* = 3318), demonstrating that the detection of mut-*KRAS* both alone and in combination with other somatic mutations (mut-*KRAS*+) is associated with poor survival irrespective of disease stage [86]. In contrast, in aPDA, mut-*KRAS* positivity has been found to have no prognostic value, but mut-*KRAS*+ detection has been found to indicate poorer outcomes. Other studies have demonstrated that the detection of *TP53* and *HER2* exon 17, with or without mut-*KRAS* variants, is significantly associated with patient outcomes [27,28]. 

### 3.3. cfDNA Epigenetic Markers

cfDNA epigenetic markers have not been studied as extensively as somatic mutations in PDA (summarized in Table 3). Henriksen et al. tested a 28-gene panel (based on a literature search and their own pre-clinical studies) by methylation-specific PCR for diagnosis and subsequent staging and prognostication (in terms of survival) in PDA [42,87]. In a study of 95 patients with PDA published in 2017, the authors showed that patients with mPDA tend to have a higher number of hypermethylated genes (10.24, of their 28-gene panel) than patients with other stage tumors (stage I, 7/09; stage II, 7; and stage III, 6.77) [88]. Through multivariable regression analysis, they identified a set of genes indicating metastasis (mPDA vs. non-metastatic PDA) and advanced disease (mPDA and stage III vs. stage I/II) with reasonable sensitivity and specificity. The same group proposed risk stratification for staging and survival (summarized in Appendix A) [89]. To do so, they used the same 28-gene panel to identify potential predictors of patient outcomes by UVA and developed a prognostic prediction model to divide the patients into four risk groups by adding an American Society of Anesthesiologists score (ASA) of 3 and performance status (Eastern Cooperative Oncology Group or ECOG) >0. This stratification was applied to the whole group (all stages), stage I/II patients, and stage IV patients. Patients with more than ten hypermethylated genes also had poor prognosis.

In a post-hoc analysis of two clinical trials of patients with mPDA (Prodige 35 and Prodige 37), ctDNA positivity (detection of methylated *HOXD8* and *POU4F1* by PCR) was found to be a poor prognostic factor for survival [91]. Multivariate analysis has identified ctDNA positivity as an independent risk factor for PFS and OS. Other factors included for OS were CA 19-9 > 1366 U/mL and ECOG of 1–2 (compared with 0). Epigenetic changes at selected CpG sites in a group of promoter genes (*p16*, *RARbeta*, *TNFRSF10C*, *APC*, *ACIN1*, *DAPK1*, *3OST2*, *BCL2*, and *CD*) were examined in whole blood samples of 30 patients with PDA (22/30 advanced stage tumors) [90]. Higher methylation levels at *TNFRSF10C* and *ACIN1* have been found to correlate with poor survival. The former is also associated with peri-neural invasion.

Cao et al. have reported the use of epigenetic changes, 5mC, and 5hmC in the diagnosis of PDA [2]. This strategy contrasts with the examination of specific promoters in prior studies. The authors started with genome-wide profiling for 5mC and 5hmC of cfDNA in the first set of healthy individuals and patients with PDA to identify differentially methylated peaks (DMPs) for 5mC and differentially hydroxymethylated peaks (DhMPs) for 5hmC between them. By testing these DMPs and DhMPs in training and validation sets, they identified 24 and 27 significantly different markers for 5mC and 5hmC, respectively. The 5hmC markers alone were able to distinguish stage I tumors from stage II, III, or IV tumors, and the 5mC and 5hmC markers together predicted tumor size (<3 cm vs. >3 cm) and perineural invasion. Notably, these markers did not predict resectability, distant metastasis, vascular invasion, or nodal involvement of the tumor. The influence of methylation status in terms of MI on PDA outcomes has been reported by Singh et al. [47]. Higher MI of *SPARC* (>0.2) and *NPTX2* (>0.34) has been found to be typical of metastatic cancer and poor survival. Advanced cancers (stage III/IV) had higher MI of *UCLH1* (>0.42) than early stages (stage I/II). Overall, epigenetic markers can aid in precise staging and prediction of outcome, but larger studies are needed before these markers can be validated for use in clinical practice.

## 4. cfDNA for POR

The current standard of care for POR in PDA is to ascertain resectability through imaging (computerized tomography or CT scans) [92]. Unfortunately, imaging provides limited information and can lead to misdiagnosis of 20% of unresectable PDAs and 36% of locally advanced or metastatic tumors as R-PDAs [93,94]. In resected tumors, the recurrence rate is very high (69–90%), and long-term survival (>10 years) is merely 4% (outcomes of key AT studies are summarized in Appendix A) [5,95,96,97,98,99,100]. Addition of NAT (chemotherapy alone with/without radiation) has not been found to be significantly associated with differences in the resection rate and survival in R-PDA, BR-PDA, and LA-PDA [101]. Insufficient tumor tissue often prevents molecular testing in early-stage PDA pre-operatively. An alternative strategy using the biomarker CA 19-9 is not always dependable (as discussed above) [11,12,13]. Inadequate POR with standard imaging (and CA 19-9) would subject patients with unresectable, LA-PDA, and mPDA (with micro-metastasis) disease to unnecessary surgical attempts/interventions and prevent them from receiving life-extending systemic therapy for 2–3 months. One possible avenue involves the use of genomic testing to identify prognostic biomarkers. However, this approach has been limited by inadequate pre-operative tissue access. Very few studies have examined POR with cfDNA testing. Some have tested post-operative blood or compared pre-operative cfDNA results with post-operative samples (Table 4). 

Pre-operative detection of *KRAS* mutation often correlates with poor outcomes, irrespective of resectability (Table 4) [71,81,84,102,103,104,105,106,107,108,109,110,111]. Alternatively, two studies have not found that pre-operative mut-*KRAS* detection is an independent risk factor for poor prognosis, but in both studies, the emergence of post-operative mut-*KRAS* was predictive of poorer outcomes [81,109]. An observational study performing cfDNA testing at the time of resection demonstrated that perioperative variations in the levels of mut-*KRAS* detected in cfDNA can be a valuable tool for prognostication after curative resection of PDA [107]. In that study, patients with pre-operative and/or intraoperative detection but no post-operative detection of circulating mutant *KRAS* had longer OS than those who were entirely *KRAS*-negative (pre/intra−/post – group) and patients with mut-*KRAS* detected in cfDNA after surgery (pre/intra+/−/post + group). Furthermore, another study reported that monitoring of post-operative levels of mut-*KRAS* in cfDNA with CA19-9 successfully predicted PFS in a panel of 25 patients who underwent curative resection [108]. The addition of other somatic mutations (*TP53*, *SMAD4*, *NRAS*, *PIK3CA*, and *STK11*) to *KRAS* has not been found to improve POR [112]. A recent meta-analysis of 38 studies evaluated the prognostic role of cfDNA in patients with PDA [86]. A subgroup analysis of five studies was performed to examine the prognostic value of pre- and post-operative total cfDNA levels after surgical resection. As expected, detectable postoperative levels of total cfDNA were associated with poor prognosis and greater risk of recurrence. However, the authors interestingly found that elevated levels of cfDNA pre-operatively were also associated with recurrence after surgical intervention. 

To our knowledge, no studies have investigated epigenetic changes in cfDNA after surgical resection in PDA. A study in patients with gastric cancer demonstrated that genes that are initially methylated pre-operatively become demethylated after surgical resection, and that this demethylation correlates with favorable prognosis [113]. Furthermore, low levels of hypermethylated cfDNA post-operatively are correlated with improved OS and PFS in gastric cancers [114]. Similarly, methylation of IKZF1 and BCAT1 detected in cfDNA from patients with colorectal cancer who underwent colectomy was found to indicate a greater risk of residual disease and recurrence [115]. Together, these studies inspire hope for similar results in PDA. 

Overall, pre-operative detection of mut-*KRAS* variants indicates poor outcomes in patients considered for resection. Prospective, randomized trials are needed to determine whether these variants could be used as surrogate markers to decide on (i) the use of NAT until the mut-*KRAS* is undetectable, (ii) continuing AT until mut-*KRAS* is negative, or (iii) closer surveillance in mut-*KRAS* positive patients after resection or completion of AT. Additionally, epigenetic markers should also be explored for POR.

## 5. Predictive Value of cfDNA Markers

Limited systemic options and poor prognosis hinder choosing the right chemotherapy combination for first-line treatment in mPDA and LA-PDA. Younger patients (<75 years) with good performance status (ECOG of 0–1) are typically treated with FOLFIRINOX based on the study population (PRODIGE 4/ACCORD 11 trial) unless patients have other contraindications to individual drugs (e.g., neuropathy for oxaliplatin), whereas other patients receive G-NP [6,7]. No reliable predictive biomarkers markers are available to guide the use of these combinations beyond these criteria.

The prevalence of available markers, such as *BRCA* 1/2 for platinum-based therapy and MSI-H for immune-checkpoint inhibitors, is very low (1–2%) [116,117]. To improve PDA outcomes, biomarkers are needed that can predict treatment response, detect treatment resistance early, and identify the benefit of rechallenging previous regimens (as in using EGFR antibodies in CRC) [118]. cfDNA testing has the potential to help in such situations. A seminal study by Leon et al. demonstrated that treatment of the underlying malignancy decreases the detectable cfDNA in patients with malignancy, thus suggesting that cfDNA has predictive value [19]. Other groups have demonstrated that cfDNA aberrations can outperform CEA in predicting the response to therapy and recurrence in patients with colorectal cancer [119].

### cfDNA Somatic Mutations

The detection of somatic mutations in cfDNA at baseline and their disappearance after initiation of therapy has been consistent with stable to partial response in prior studies [72,78,81,120]. In the same studies, continued detection and/or re-emergence of mutations have been found to be reliable biomarkers for disease progression. In one study, PFS was found to be 5-fold greater in mut-*KRAS* negative patients than in those who continued to test positive after 4–8 weeks (PFS of 248.5 in positive vs. 50 days negative, *p* < 0.001) [78]. Watanabe et al. showed that its (mut-*KRAS*) detection after 6 months significantly correlates with unfavorable therapeutic responses to first line chemotherapy [81]. In a different study, *TP53* and/or *KRAS* clearance, compared with a lack of clearance, have been associated with better patient response and outcomes [27].

In a study published in 2017, blood samples were collected from 27 patients before the first dose of chemotherapy (FOLFIRINOX or G-NP), on day 15 of the first cycle, and at subsequent clinic visits [69]. The mut-*KRAS* dynamics (changes in concentrations) in the samples collected 15 days after the first dose were indicative of treatment response and survival. Increased mut-*KRAS* levels correlated well with poor survival and ineffective therapy (increase vs. no increase or reduction, median PFS of 2.5 vs. 7.5 months, *p* = 0.03; median OS of 6.5 vs. 11.5 months, *p* = 0.009). Another study tracked mut-*KRAS* in cfDNA from 54 patients with aPDA, most of whom (37/54) received first-line gemcitabine (Gem)-based therapy (Gem-only, or Gem with erlotinib, axitinib, or everolimus) [120]. An early decrease in mut-*KRAS* concentrations 14 days after chemotherapy was associated with a radiographic response at 6 months. A key finding was that cfDNA can be used to predict response to chemotherapy after one cycle and can therefore be used evaluate the effectiveness (risk of adverse events vs. benefit) of first-line treatment. Cheng et al. demonstrated the predictive value of *KRAS* 12V in patients treated with G-NP [28].

A prospective observational study in 38 patients with PDA receiving first-line FOLFIRINOX demonstrated similar results [32]. In that study, cfDNA profiling was performed on blood samples drawn before treatment initiation and at the time of response assessment. Changes in cfDNA MAF directly correlated with treatment response and disease burden. In the described cases, apart from *KRAS*, the MAFs of other mutations that correlated with patient response in serial testing were for *TP53*, *GNAS*, *SF3B1*, *CDK2NA*, *KEAP*-1, *BCL11A*, *PBRM1*, *RB1*, *ARID1A*, and *FGFR1*. In an NGS-based study (*n* = 17 for detectable for detectable cfDNA), patients with low MAF and copy number variants were found to have either a complete or partial response to chemotherapy [121]. Systemic therapies received by patients in this group included FOLFIRINOX, G-NP, Gem-only, Gem-erlotinib, and capecitabine/oxaliplatin.

Monitoring levels of mut-*KRAS* cfDNA during neoadjuvant chemotherapy have also been demonstrated to predict which patients may eventually be candidates for surgical resection [66]. In a study of 34 patients with PDA (18/34 and 16/34 received G-NP and FOLFIRINOX, respectively; 26 had chemo-radiation, 22 with capecitabine, and 4 with Gem), 71% of patients who underwent resection showed a statically significant decrease in levels of mut-*KRAS* cfDNA compared with pre-treatment. In contrast, among the patients who did not undergo resection, 94% had no reduction in, or increased, mut-*KRAS* levels.

PDAs with homologous recombination deficiency (HRD) or DNA damage repair (DDR) deficiency are known to have a good outcome as they respond to platinum agents better (than PDAs without them) [122,123]. cfDNA studies that specifically looked at the prevalence of such rare mutations (somatic and germline) are limited, but commercially available panels cover most of them [124]. Detecting pathogenic mutations in DDR genes such as *BRCA 1* or *2* early in diagnosis can help the treating physician consider platinum therapy (FOLFIRINOX or gemcitabine/cisplatin) in the first line and poly (ADP-ribose) polymerase (PARP) inhibitors (PARPi) for maintenance [125,126,127,128]. A retrospective study (*n* = 11, 9 PDA) reported a reasonable response (complete response, partial response, and stable disease was noted in two, one, and two PDA patients, respectively) of chemotherapy-refractive HRD-positive PDA with ipilimumab/nivolumab [129]. Therefore, even though HRD-positive and DDR-positive PDAs are rare, efforts to identify them are encouraged, given their impact on treatment selection and, ultimately, the outcome.

To our knowledge, no studies have directly investigated aberrations in cfDNA methylation in response to chemotherapy in PDA. Studies have demonstrated that concentrations of methylated cfDNA at *RASSF1A* correlate with the response to neoadjuvant therapy in patients with breast cancer: patients with a decline in methylated *RASSF1A* cfDNA compared with pre-treatment often responded to neoadjuvant therapy [130]. A similar study in non-small cell lung cancer demonstrated that a decrease in methylated cfDNA in a ten-locus gene set predicts response to therapy and OS [131]. Furthermore, a study in patients with colorectal cancer demonstrated that treatment with regorafenib increases the total cfDNA; however, the concentration of total methylated cfDNA decreases [132]. This decrease in methylated cfDNA correlates with therapeutic response to treatment. Interestingly, the authors have also demonstrated that high methylated cfDNA levels during treatment correlate with minimal regorafenib response. Future studies are needed to assess the dynamics of methylation of cfDNA in response to therapy in patients with PDA to determine signatures and expression profiles that can predict response to therapy.

## 6. Conclusions

Efforts to improve outcomes in PDA should start with early detection, better prognostication methods to overcome the drawbacks of standard imaging and CA 19-9, dependable predictive biomarkers, and superior POR systems. Prognostic and predictive biomarkers can be exploited to develop newer targets or agents to treat PDA. The latter would help clinicians choose the optimal first-line agent for PT or NAT. The evidence provided above provides hope that cfDNA might provide an ideal source of biomarkers. The current evidence of cfDNA in PDA can be summarized as indicated in Figure 1.

Prior studies have demonstrated the feasibility of detecting genetic alterations (cfDNA) in bodily fluids (such as blood, pancreatic juice, and bile) with available technology (ddPCR or NGS) in PDA. cfDNA from blood, compared with other sources, provides an ideal non-invasive alternative to study such alterations, particularly for monitoring disease. Mutations, methylation alterations, and other genetic events such as microsatellite aberrations and loss of heterozygosity identified in cfDNA may serve as markers of malignant transformation and as biomarkers for the early stages of PDAs [2,32]. Substantial evidence supporting the use any markers for the screening and diagnosis of PDA is lacking. Nonetheless, these markers may help confirm the diagnosis if EUS-FNA is not conclusive, or if a patient has carcinoma of unknown origin with non-specific pathology.

NGS-based cfDNA mutation testing could be offered for patients with PDA to identify targetable mutations if tumor tissue is unavailable or insufficient. The reasonable concordance between tumor and cfDNA testing reported in earlier studies suggests that this method provides a reliable alternative tissue mutation profiling. Notably, the top ten mutations identified in the cfDNA of patients with PDA are currently not targetable. Epigenetic marker identification is not accessible outside experimental settings and currently has no demonstrated therapeutic value in PDA. Non-specific demethylating agents such as decitabine and azacitidine are under investigation in phase I or I/II clinical trials, which have not required specific epigenetic markers (in tissue or cfDNA) for patient enrollment [133].

cfDNA has a no significant utility as a biomarker in current clinical practice. Most studies focused on mut-*KRAS* variants (detected by PCR) before the era of NGS-based studies. Although some studies have demonstrated that mut-*KRAS* detection, concentration, and MAF can provide insight into outcomes and treatment responses in patients with PDA, other larger studies and meta-analyses have indicated the importance of other mutations [27,28,75,77,85,86]. Efforts should be made to identify those in cfDNAs and develop panels of specific mutations instead of relying on only mut-*KRAS* variants in future studies. cfDNA epigenetic studies to date have not addressed prognosis or treatment response sufficiently to support their use. High prevalence of epigenetic markers among the patients with PDA compared to HC gives them a distinct advantage over somatic mutations for diagnosis and screening. As additional methylation signatures for diagnosis are identified, their prognostic and predictive value must be assessed in future studies. All the facets of epigenetic markers, such as detection (non-specific markers such as 5mC and 5hmC and specific promoters), concentrations, and MI, should be considered in future investigations. Prior cfDNA testing studies have not tackled POR adequately, including the detection of micrometastasis, risk of recurrence, and need/duration of NAT, particularly in BR-PDA. Finally, cfDNA testing cannot aid in choosing the ideal first-line therapy option between FOLFIRINOX and G-NP or in identifying early resistance to therapy. 

In conclusion, cfDNA testing appears promising in PDA management, but its utility is mainly restricted to identifying targetable mutations in current clinical practice. It is not ready for diagnosis, screening, prognostication, prediction treatment response, monitoring of disease burden, and POR. The low prevalence of mutations in cfDNA hinders research and the drawing of reliable conclusions. Studies have suggested that epigenetic biomarkers may aid in all the critical areas discussed above. Future studies should further pursue epigenetic biomarkers to identify factors that may be exploited for therapeutic purposes.

## Figures and Tables

**Figure 1 cancers-14-03453-f001:**
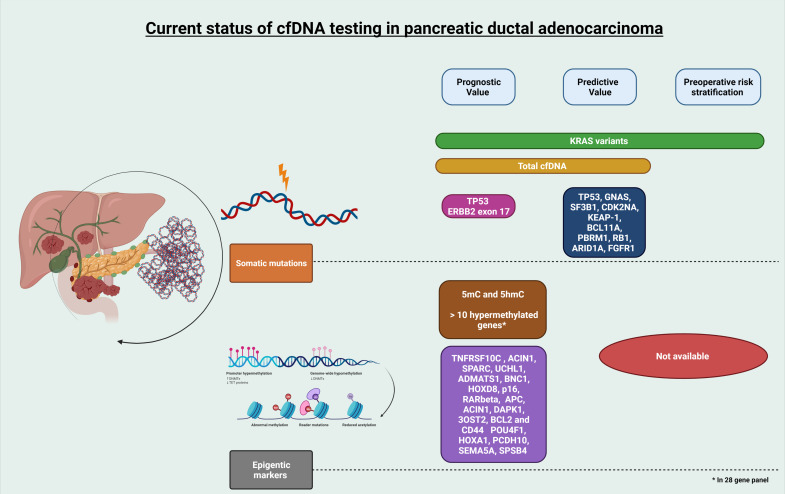
cfDNA somatic mutations and epigenetic markers in pancreatic ductal adenocarcinoma. Summary of specific somatic mutations and epigenetic markers that can be detected in cfDNA which have been shown to confer prognostic, predicative, and preoperative risk stratification value in patients with pancreatic ductal adenocarcinoma.

**Table 1 cancers-14-03453-t001:** cfDNA methylation marker studies to diagnose pancreatic ductal adenocarcinoma.

Study	Study Population	Technology Used	Target	Sensitivity and Specificity, Respectively
Cao, Wei, Hu, He, Zhang, Xia, Tu, Yuan, Guo, Liu, Xie, and Li [2]	HC vs. PDA	Methylome sequencing	24-feature 5mC and 27-feature 5hmC model	5mC + 5hmC—93.8% and 95.5% (AUC of 0.99) 5mC alone—87% and 82% (AUC of 0.97) 5hmC alone—78–85% and 99–100% (AUC of 0.99–0.96)
Melnikov et al. [38]	HC vs. PDA	Microarray	* CCND2, PLAU, SOCS1, THBS, and VHL	76% and 59%
Liggett et al. [39]	HC vs. PDA vs. CP	Same microarray as Melnikov et al., 2009	14 promoters for PDA vs. CP 8 for HC and CP	91.2% and 90.8% (PDA vs. CP) 82% and 78%
Yi et al. [40]	PDA vs. PanIN or pancreatitis	PCR	BNC1 and ADAMTS1	Both—81% and 85% BNC1—79% and 89% ADAMTS1—48% and 92%
Eissa et al. [41]	PDA vs. noncancer	PCR	BNC1 and ADAMTS1	Combined AUC of 0.95 *ADAMTS1*—87.2% and 95.8% (AUC = 0.91; 95%) *BNC1*—64.1% and 93.7% (AUC = 0.79) Both—97.3% and 91.6% (AUC = 0.95)
Henriksen et al. [42]	PDA vs. HC + CP + acute pancreatitis	PCR	BMP3, RASSF1A, BNC1, MESTv2, TFPI2, APDA, SFRP1, and SFRP2 With age >65 years	76% and 83% (0.86)
Guler et al. [43]	PDA vs. no PDA	CHIP seq	** Gene set of top 65% differentially hydroxymethylated genes	Training set AUC = 0.92 Validation set 1 AUC = 0.921 Validation set 2 AUC = 0.943
Li et al. [44]	HC vs. PDA	MeDIP-seq	TRIM73, FAM150A, EPB41L3, SIX3, MIR663, MAPT, LOC100128977, and LOC100130148	93.2% and 95%
Ying et al. [45]	HC vs. PDA	PCR	ADAMTS1, BNC1, LRFN5, and PXDN	100% and 90%
Manoochehri et al. [46]	HC vs. PDA	ddPCR	SST	100% and 89%
Singh et al. [47]	HC vs. CP vs. HC	PCR	SPRC, UCHL1, NPTX2, and PENK	HC vs. CP + PDA, MI of all 4 are increased HC vs. CP, MI of UCLH1, PENK, NPTX2 increased CP vs. PDA, MI of SPARC, and NPTX2 increased CP vs. early-stage PDA, MI of SPARC increased
Shinjo et al. [48]	PDA vs. HC	MBD-ddPCR	ADAMTS1, HOXA1, PCDH10, SEMA5A, and SPSB4	Panel of 5 genes with 49% and 86% 1 of 5 in 49% of PDA Panel of 5 genes + *KRAS* mutation in cfDNA with 68% and 86%
Fujimoto et al. [49]	PDA vs. benign disease and HC	PCR	RUNX3	RUNX3 alone: 50.9% and 93.5% RUNX3 combined with CA19-9: 85.5% and 93.5% for all stages and 78% for stage I
Kandimalla et al. [50]	PDA vs. HC	Genome-wide DNA methylation sequencing	EpiPanGiDx	Predictive value of 85%
Vrba et al. [51]	PDA vs. benign cyst	PCR	10-promoter panel in mPDA	100% and 95% (AUC of 0.999)
Li et al. [52]	PDA vs. PanIN benign tumors and pancreatitis	PCR	BNC1 SEPT9 in Stage I and II	Combined—65% and 87% BCN1—50.9% and 88.7% SEPT9—36.8% and 96.2% Combined + CA 19-9 vs. CA19-9 alone—86% vs. 61.4% and 81.1% vs. 90.6% Individually they have low CT compared to HC and benign disease
Melson et al. [53]	PDA vs. HC	PCR ^#^	VHL, MYF3, TMS, GPC3, and SRBC	80% and 66% (AUC = 0.848)
Park et al. [54]	PDA vs. CP	PCR	NPTX2	80% and 76%
Park et al. [55]	PDA vs. HC	PCR	P16	Higher methylation in PDA than HC (86.7 ± 29.8 vs. 33.3 ± 0.00, *p* = 0.016)

PDA—pancreatic ductal adenocarcinoma; mPDA—metastatic PDA; AUC—area under the receiver operating characteristic curve; PCR—polymerase chain reaction, ddPCR—digital droplet PCR; ChIP-Seq—chromatin immunoprecipitation sequencing; MeDIP-Seq—methylated DNA immunoprecipitation sequencing; MBD-ddPCR—enrichment of methyl-CpGbinding (MBD) protein count-followed by ddPCR; CT—cycle thresholds; * panel of reduced methylation; ** combination of hypo and hyper methylated genes; ^#^ methylation sensitive restriction enzyme and multiplex PCR.

**Table 2 cancers-14-03453-t002:** Summary of studies determining prognostic value KRAS mutations in cfDNA.

*KRAS* Variants	Technique	Findings
G12A, G12C, G12D, G12R, G12S, G12V, and G13D [64]	ddPCR	Detection of mutant *KRAS* is associated with poor OS (197 days vs. 60 days, HR = 2.8, *p* = 0.018).
G12D, G12V, and G12R [65]	ddPCR	*KRAS* mutation at G12V conferred poorer OS compared to WT (*p* < 0.01). No significance effect of *KRAS* G12D mutation.
G12A, G12C, G12D, G12R, G12S, G12V, and G13D [66]	ddPCR	MAF ≥ 5% of any variant was a poor predictor of PFS (HR = 2.28; 95% CI: 1.18–4.40; *p* = 0.014) and OS (HR = 3.46; 95% CI: 1.40–8.50; *p* = 0.007). MAF peak above 1% was significantly associated with radiologic progression (*p* = 0.0003).
*KRAS* Codon 12 mutations [67]	PCR	Detection of *KRAS* mutations conferred shorter OS compared to WT (3.9 months vs. 10.2 months, *p* < 0.001). Mutational burden could significantly correlate with TNM tumor staging (*p* = 0.033) and liver metastasis (*p* = 0.014). *KRAS* mutations were a negative prognostic factor for survival (HR = 7.39; 95% CI: 3.69–14.89).
G12V and G12D [68]	ddPCR	G12V conferred poor OS *(p* = 0.001). G12D conferred poor OS (*p* = 0.044).
G12D, G12V, G12R, and G13D [69]	ddPCR	Increased mutational burden conferred poor PFS (2.5 vs. 7.5 months, *p* = 0.03) and OS (6.5 vs. 11.5 months, *p* = 0.009).
G12D, G12V, G12R, and G12C [70]	NGS	G12R mutation conferred favorable OS compared to WT (20.4 vs. 14.5 m, HR = 0.67 (95% CI: 0.47–0.93), *p* = 0.0215) and PFS on first-line therapy (12.2 vs. 6.8 m, HR 0.60 (95% CI 0.40–0.85), *p* = 0.004).
G12A, G12C, G12D, G12R, G12S, G12V, and G13D [71]	ddPCR	*KRAS* mutations concentration >0.165 copies/L had worse OS median fractional abundance (>0.415%.)
G12V, G12D, and G12R in codon 12 *KRAS* [34]	ddPCR	mOS was significantly shorter in patients with *KRAS* mutant (276 days) compared with patients with WT *KRAS* (413 days) from cfDNA samples (*p* = 0.02). mOS was significantly shorter only in G12V variants compared to other *KRAS* mutants (219 days vs. 410 days, *p* = 0.006).
G12D, G12R, G12V, Q61H, Q61R, and A59G [72]	BEAM-PCR	Overall response rate, disease control rate, mPFS, and mOS were higher in patients without detectable *KRAS* mutations (48% vs. 28%, 81% vs. 69%, 8.8 vs. 5.3 months, and 18.2 vs. 6.6 months, respectively).
*KRAS* codon 12 [73]	ddPCR	Patients with WT *KRAS* had better OS than mutant *KRAS* patients (10.6 months vs. 5.6 months, *p* < 0.05). Patients with *KRAS* mutation and copy number gain had the worst prognosis with a mOS of 2.5 months (*p* ≤ 0.0001).
*KRAS* exon 12 [74]	PCR	Undetectable mutant KRAS conferred favorable OS (8 vs. 37.5 months from diagnosis, *p* < 0.004).
*KRAS* mutations: not specified [75]	DNA-based Ion-Torrent NGS assays (ClearID)	Presence of *KRAS* mutations in cfDNA was associated with reduced mOS (54% in mutation-positive versus 90% in mutation-negative, *p* < 0.05).
*KRAS* codon 12 and 13 [76]	PCR	Patients with *KRAS* mutations detected in cfDNA had significantly lower mPFS (1.8 vs. 4.6 months, *p* = 0.014) and mOS (3.0 vs. 10.5 months, *p* = 0.003) than those without detected plasma *KRAS* mutations.
G12A, G12C, G12D, G12V, G12R, G12S, and G13D [77]	ddPCR and NGS amplicon panel	Detectable mutant *KRAS* cfDNA was associated with poor OS (3.2 vs. 8.4 months, *p* = 0.005).
*KRAS* G12/13 mutations and *KRAS* Q61K [78]	ddPCR	Detectable mutant *KRAS* cfDNA was associated with poor PFS (308.5 vs. 168 days, *p* = 0.07).
G12D, G12R, G12V, and G13D [79]	ddPCR	Higher concentration with advanced stages (*p* = 0.0129).
*RAS* mutation (*KRAS*/*NRAS* codons 12, 13, 59, 61, 117, and 146) [80]	BEAM-PCR	Higher *RAS* MAF was associated with poor OS (142 vs. 310 days, *p* = 0.0261) and PFS (85 versus 175 days; *p* = 0.0556).
G12V, G12D, G12R, and Q61H [81]	ddPCR	The mOS of patients with detectable mutant *KRAS* cfDNA was shorter (15.8 months vs. 33.7 months; *p* < 0.05)
G12D, G12V, G12R, G13D [32]	Digital PCR and NGS	Patients with multiple liver metastasis and poor mOS had higher mutant *KRAS* cfDNA MAF compared to those with fewer lesions (*p* < 0.05).

PCR—polymerase chain reaction; ddPCR—digital droplet PCR; BEAM-PCR—beads, emulsion, amplification, and magnetics PCR; NGS—next generation sequencing; WT—wild type; MAF—mutant allelic fraction; HR—hazards ratio; CI—confidence interval; mOS—median overall survival; mPFS—median progression-free survival.

**Table 3 cancers-14-03453-t003:** Summary of studies examining epigenetic markers in cfDNA relating to prognosis of patients with PDA.

Genes Studied	Comparison	Findings
5mC and 5hmC pan-sequencing [2]	Identify DMPs for 5mC and DhMPs for 5hmC	5mC: No difference between resectable vs. unresectable PDA. 5hmC: Significant different between stage I vs. II/III/IV; no difference between resectable vs. unresectable. 5mC + 5hmC: Higher in tumor size <3 cm (vs. >3 cm) and PNI; no difference between resectable vs. unresectable; no significant differences in vascular invasion or positive lymph node metastasis were found in resectable PDA patients.
SPARC UCLH1 PENK NPTX2 [47]	Low vs. high methylation index	SPARC: Higher in stage IV and poor survival (3 vs. 6 m); Lower in resectable (*p* = 0.02). NPTX2: Higher in stage IV, met dz and poor survival (3 vs. 9 m) (*p* = 0.04). UCLH1: Higher in stage III/IV vs. I/II (*p* = 0.034).
28-gene panel for staging [88]	Number of methylated genes	Stage I: 7.09 (95% CI: 5.51–8.66). Stage II: 7.00 (95% CI: 5.93–8.07). Stage III: 6.77 (95% CI: 5.08–8.46). Stage IV: 10.24 (95% CI: 8.88–11.60). The number of methylated genes at stage IV was significantly higher compared to stage I/II/III PDA (*p* = 0.0002).
	Specific promoters	The prediction model (SEPT9v2, SST, ALX4, CDKN2B, HIC1, MLH1, NEUROG1, and BNC1) enabled the differentiation of stage IV from stage I-III disease (AUC of 0.87 (cut point: 0.55); sensitivity of 74%, specificity of 87%)). Model (MLH1, SEPT9v2, BNC1, ALX4, CDKN2B, NEUROG1, WNT5A, and TFPI2) enabled the differentiation of stage I-II from stage III-IV disease (AUC of 0.82 (cut point: 0.66); sensitivity of 73%, specificity of 80%)).
Same panel as above [89]	Number of methylated genes Risk stratification based on ASA= 3 and methylation of GSTP1, SFRP2, BNC1, SFRP1, TFPI2, and WNT5A	Patients with more than 10 hypermethylated genes had an HR of 2.03 (95% CI: 1.15–3.57). Total group (all stages of tumor) HR compared to group 1: Risk group 2: HR 2.65 (95% CI: 1.24–5.66); Risk group 3: HR of 4.34 (95% CI: 1.98–9.51); Risk group 4: HR of 21.19 (95% CI: 8.61–52.15). Stage I–II (ASA = 3, SFRP2, and MESTv2): Risk group 2: HR of 4.83 (95% CI: 2.01–11.57); Risk group 3: HR of 9.12 (95% CI: 2.18–38.25); Risk group 4: HR of 70.90 (95% CI: 12.63–397.96). Stage IV (BMP3, NPTX2, SFRP1, and MGMT): Risk group 2: HR of 5.23 (95% CI: 2.13–12.82).
ADAMTS1, HOXA1, PCDH10, SEMA5A, and SPSB4 ± *KRAS* mutations [48]	Positive vs. negative	Large tumor size and higher frequency of liver metastatic disease in cfDNA positive patients
p16, RARbeta, TNFRSF10C, APC, ACIN1, DAPK1, 3OST2, BCL2, and CD44 [90]	Methylation levels in CpG promoter regions	The highest tertile of methylation of ACIN1 was associated with shorter survival compared to the middle and the lowest tertile group (13 months vs. 17 months). Highest tertile of TNFRSF10C was associated with shorter survival compared to the middle and the lowest tertile group (OS, 13 months vs. 22 months). TNFRSF10C SN1 methylation was significantly associated with PNI (OR = 0.088).
HOXD8 and POU4F1 [91]	Detection	Median PFS and OS were 5.3 and 8.2 months in ctDNA-positive and 6.2 and 12.6 months in ctDNA-negative patients, respectively. ctDNA positivity was more often associated with young age, high CA19-9 level, and neutrophils lymphocytes ratio. ctDNA was confirmed as an independent prognostic marker for PFS (HR = 1.5, CI 95%: [1.03–2.18], *p* = 0.034) and OS (HR = 1.62, CI 95%: [1.05–2.5], *p* = 0.029).

5mC—methylated methylcytosine; 5hmC—5-hydroxymethyl cytosine; DMPs—differentially methylated peaks; DhMPs—differentially hydroxymethylated peaks; PNI—perineural invasion; ASA—American Society of Anesthesiologists score; PFS—progression-free survival; OS—overall survival; HR—hazards ratio; CI—confidence interval; OR—odds ratio.

**Table 4 cancers-14-03453-t004:** Preoperative risk stratification of pancreatic ductal adenocarcinoma.

Study	*KRAS* Variants Tested	Impact of Preop Detection	Impact of Postop Detection
Groot et al., 2019 [102]	G12V/12D/12R/Q61H	Significant	Persistent—significant
Lee et al., 2019 [103]	Codons 12/13/61		Significant
Yamaguchi et al., 2021 [104]	G12/12V/12R		Not significant
Guo et al., 2020 [105]	G12D		Not studied
Hadano et al., 2016 [106]	G12V/12D/12R		
Kim et al., 2018 [71]	G12A/12C/12D/12R/12S/12V/13D		
Hipp et al., 2021 [107]	G12D/12V/12R/12C	Not Significant	Significant
Hussung et al., 2021 [108]	Codons 12/13/61		Significant
Nakano et al., 2018 [109]	Codons 12/13		Conversion from wild type to mutation—significant
Wantanabe et al., 2019 [81]	G12V/12D/12R/Q61H		Emergence—significant
Sausen et al., 2015 [110]	G12V/12D/12R/12V/12C/13D	Not studied	Significant

Preop—preoperative; Postop—post operative.

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
