# Peer review of "Is Cell-Free DNA Testing in Pancreatic Ductal Adenocarcinoma Ready for Prime Time?"

_cancers, 2022, doi:10.3390/cancers14143453_

Round 1
Reviewer 1 Report
In this reviews Authors provide an extended evaluation of the role of cfDA in PDAC, with a particular to the role of somatic mutations and methylation changes, with interesting focus on predictive and prognostic role of the test in the management of PDAC.
The topic is very appeal in this area, because at this moment there are no manageable and simple tests with prognostic and predictive value for PDAC, except for ca 19.9, which is not strongly specific. Moreover, the review underlines the importance of identifying new diagnostic tools that can replace the not always feasible tissue biopsy
The review is very complete and provides an extensive excursus about detection techniques, literature and relation with stages of disease and outcomes. It provide a very updated review of latest data.
I suggest to add some evidences about the role of liquid biopsy in the diagnostic of subgroup of HRD pancreatic cancers, and the potential implication of early detection of DDR tumors for treatment.
Conclusions are pragmatic and well balanced. References are reported an fully complete.
I suggest to reduce the size of the tables, and to make the image bigger
Author Response
We appreciate the valuable input that made out review better. Please find the responses to the suggestions in the attached document
Thanks

Reviewer 2 Report
In this review, Sheel et al. provide a comprehensive overview of the published literature about the role of cell-free DNA in pancreatic cancer, an interesting new alternative/adjunct to traditional diagnosis of PDAC. This is a very timely topic and of interest to a broad readership. The review has a good structure with sufficient introduction and conclusion.
Below a few minor suggestions:
1. Please revise and shorten the introduction of the article. The very first sentence seems to be missing an “a” prior to “fatal disease”. Reporting the relative survival seems unusual and so does the “rate of deaths to new cases” (line 46)
2. Please double check the article for any typos, for example line 65 “area” should be “are”
3. Since the current management of pancreas cancer is not the main focus of the article, section 1.1 can also be shortened and simplified
4. Please clarify line 204-206: The concordance rate, i.e. the rate that the mutations were detected in exclusively blood and tissue samples, was just 32%, 30% and 38%, respectively. I am not sure concordance is defined correctly here
Author Response

(The authors gave the same response as above.)
